# Social and Economic Influence of Sustainable Development: The Case of Al-Mouj, Muscat, Oman

**Eman Hanye Mohamed Nasr [1,2]**, **Aisha Mohammed Al Shehhi [2]** and **Mohamed Ali Mohamed Khalil [1,3,*]**

[1] Department of Architectural Engineering, Faculty of Engineering, Mansoura University, Mansoura 35516, Egypt; eman.h@mans.edu.eg or eman.n@gcet.edu.om

[2] Department of Urban Planning and Environmental Management, Global College of Engineering and Technology, Muscat 112, Oman; 202011128@gcet.edu.om

[3] Department of Architecture and Interior Design, Scientific College of Design, Muscat 114, Oman

[*] Correspondence: m_khalil@mans.edu.eg or m.khalil@scd.edu.om; Tel.: +968-71722712

## Abstract

The sultanate of Oman has joined other nations in promoting sustainability, guided by Oman Vision 2040 and the Oman National Spatial Strategy. Oman now focuses on developing more human-centered cities, enhancing community well-being, boosting the local economy, and increasing investments. This study addresses a research gap by examining the social and economic impact of the sustainable neighborhood "Al-Mouj" on the nearby urban area "Al-Mawaleh North" to maximize sustainability benefits. It analyzes how a sustainable neighborhood influences the economy, society, quality of life, and overall well-being. The study also identifies key factors driving the growth of sustainable practices in society and the economy. It has four main objectives in terms of answering the research question, primarily through surveys of community members and business owners, and by analyzing land use development around Al-Mouj. Data collection methods include literature review, case study, questionnaires, and interviews. Data analysis employs spatial, statistical, and thematic techniques. Responses from 515 participants are examined to ensure reliable results. Ethnographic methods are used to gain insights from open-ended questionnaire responses and interviews. The results confirm that Al-Mouj's mixed-use development and sustainability features positively influence mental and physical health and stimulate economic activity within the local community. This study provides decision-makers and urban planners valuable insights into sustainable neighborhoods' social and economic impacts when developed as open communities. It highlights the challenges of following international NSAT standards, which do not fully address local concerns.

**Keywords:** sustainable development; social economic impact; social cohesion; well-being; local economy; Oman

## 1. Introduction

The 1987 Brundtland Report of the United Nations defined sustainable development as meeting the needs of current generations without compromising the ability of future generations to meet theirs [1]. This concept is far from selfish; it is a way of life that shows future generations that dreams can be achieved [2]. The report emphasized the link between humanity's hopes for a better life and nature's limits. Over time, the concept has been expanded to include the three pillars of sustainability: social, economic, and environmental [3]. The idea of community plays a vital role in this transition, where all parties are encouraged to collaborate [4]. In recent years, awareness of sustainability

issues has grown, as sustainability covers different aspects of resource use and society [5,6]. Because of this awareness, Oman Vision 2040 aims to promote sustainable development. Oman is among the countries that have committed to implementing the United Nations Sustainable Development Goals 2030 in all national plans and strategies [7]. Oman's future vision also aims to develop sustainable cities where economic growth and social justice flourish in urban and rural areas [8].

The sustainability of cities has become crucial in tackling environmental, social, and economic challenges, especially as it is projected that 60% of the world's population will live in cities by 2050 [9]. Furthermore, cities are the hub of human activity, and most environmental degradation and consumption occur within urban areas [10]. Nevertheless, cities and their residents can play a crucial role in reaching global sustainability [11]. This argument is based on the idea that sustainable development primarily emphasizes the connection between human activity and the environment [12]. This prompted Oman to develop the Oman National Spatial Strategy, which aims to create livable, resilient cities and urban communities while preserving the Omani identity through responsible development and sustainable use of natural resources [7].

Over the last century, urban planning theories have developed, with Howard's 1898 "Garden City" theory [13] regarded as the foundation of modern sustainable urban development. Howard criticized Britain's overcrowded industrial cities and proposed the 'Garden City' to combine urban and rural benefits, promote community, enhance sustainability, and generate economic opportunities. His focus on spatial layouts and city features like housing and neighborhoods represented a significant advance in urban planning [13]. In 1929, Clarence Perry built upon Howard's theory to create his neighborhood concept, highlighting the importance of central services, accessibility, and amenities such as schools at neighborhood centers. He emphasized social interaction and how neighborhood quality influences user experience. Perry's ideas contributed to enhanced community livability, especially as the rise of cars changed urban social interactions [14]. These theories establish a foundation for a sustainable neighborhood.

A neighborhood is a crucial part of a city, being the most livable area where residents spend much of their time building social connections, contributing to the creation of more livable urban spaces [15]. It is no longer surprising that planners and decision-makers increasingly view neighborhoods as the fundamental units of cities and the closest environmental, social, and economic contexts for residents, where sustainability can be effectively assessed [10]. The neighborhood scale is the fundamental urban development unit [16] and the baseline for assessing social, economic, and institutional sustainability aspects [10,17]. The concept of neighborhood sustainability can be seen as a process where achievements become everyday parts of life [18]. This idea is valuable because it highlights citizens' daily efforts to promote sustainability instead of relying on centralized institutions [19]. This encourages countries that plan for sustainable development to create sustainable neighborhoods as a key part of their overall development efforts. Like other countries, Oman planned to have sustainable neighborhoods that ensure a high quality of life, work, and leisure.

Social and economic sustainability goes beyond survival, aiming to improve residents' well-being and make neighborhoods enjoyable places to live, work, or visit. This is essential for true sustainability, as popular destinations draw people, investment, and renewal [20]. Social sustainability involves meeting basic needs, building social capital, and preserving community traits while promoting social justice and equality [21]. Economic sustainability involves a society, economy, and environment with minimal risks to long-term city development. It considers economic sustainability as a complex concept that includes economic growth supporting future well-being while safeguarding the rights of current

residents [22,23]. Since sustainability depends on the context, the concept of a sustainable neighborhood differs. For some, it means access to resources; for others, it emphasizes resource conservation. Research often assesses physical factors like density, land use mix, connectivity, and green spaces, which are linked to social, environmental, and economic impacts [24,25].

Planners and environmentalists began developing tools for neighborhood-scale sustainability assessments in the early 2000 s [26]. The rapid growth of cities, environmental concerns, and the demand for sustainable neighborhoods have led to Neighborhood Sustainability Assessment Tools (NSATs) like LEED-ND and BREEAM-Communities [27]. NSATs are tools designed to assess a neighborhood's progress toward sustainability objectives [28]. The NSAT used in this study is the Building Research Establishment Environmental Assessment Method (BREEAM), developed in 1990 by the Building Research Establishment (BRE) in the United Kingdom. BREEAM is the first and leading sustainability assessment for the built environment, encouraging sustainable master-planning, infrastructure, and buildings. It promotes environmental, social, and economic sustainability [29]. As part of Oman's efforts to promote sustainable development, the country has established certified neighborhoods that can serve as the foundation for sustainable cities like Al-Mouj Muscat, the Ras Al Hamra development project, and several other projects under construction.

NSATs cannot fully address various aspects of sustainability, mainly social and economic factors. This might reinforce the idea that sustainability is achieved by focusing only on specific areas instead of considering the complex relationships between building and urban environments [30]. This encourages reconsidering the spatial boundaries of sustainable neighborhoods' impact. Many researchers have found that social and economic impact studies help urban planners and decision-makers make better choices that benefit society [31,32]. It is crucial to analyze how a sustainable neighborhood can influence the social and economic aspects of the surrounding area, especially for a certified neighborhood with defined boundaries, which this study aims to investigate. This is particularly relevant since NSAT does not examine this impact to maximize the benefits of sustainable development, especially in Oman, where there is limited practice of sustainable neighborhoods and efforts to develop more successful models.

## 2. Literature Review

A systematic review was conducted to identify international and regional evidence on the history of examining urban development's social and economic impacts and whether successful criteria have been tested for assessing the best measures of these impacts for use in this study. Additionally, the review aimed to identify gaps in the literature regarding investigating the social and economic impacts of sustainable development. The databases consulted included ScienceDirect and ProQuest platforms, searching for papers indexed in Scopus or Web of Science. The inclusion or exclusion criteria were based on whether the literature successfully examined one of urban development's social or economic impacts and whether it identified whether the impact was positive or negative.

Hubbard (1995) [33] highlighted that improved urban design can promote local economic growth. It is essential to reevaluate how urban design impacts economic recovery. While urban design policies can enhance the local economy, they might also overlook urgent social issues [33]. French and others (2014) highlighted how urban design can promote social interaction [34]. Forrest and Kearns (2001) [35] emphasized a renewed focus on local social relationships, especially those involving social capital. They highlighted that neighborhoods are crucial in shaping social identity and life opportunities. They pointed out that society faces a new crisis in social cohesion and described its main features. The

authors also discussed how modern residential neighborhoods connect to broader debates on social capital, stressing the link between social cohesion and social capital. Furthermore, they demonstrated how social capital can be broken down into core areas suitable for policy efforts at the neighborhood level. They also examined strategies for integrating social cohesion and social capital into research [35].

In 2003, Leyden examined how neighborhood design influences social capital levels. Using a household survey, the study measured residents' social capital in areas ranging from traditional, mixed-use, pedestrian-friendly layouts to modern, car-focused suburban neighborhoods in Galway, Ireland. The results show that residents of walkable, mixed-use areas tend to have higher social capital than those in car-centric suburbs. Specifically, walkable neighborhood respondents were likelier to know their neighbors, participate in political activities, trust others, and engage socially. Overall, walkable and mixed-use neighborhood designs can foster the development of social capital [36]. Neighborhood studies have attracted researchers from diverse perspectives. Choguill (2008) [37] argued that a city cannot be sustainable if its neighborhood, one of its core components, is not sustainable. He aimed to create criteria to evaluate neighborhood sustainability and to explore the concept within neighborhood development. Choguill reviewed existing criteria for measuring sustainability and listed key features of sustainable neighborhoods, such as walkability, proximity to services, parks and green spaces, community centers, mixed density, and mixed-use buildings. The study highlighted that neighborhood planning theories have evolved to promote sustainability [37].

Dempsey, Brown, and Bramley (2012) [38] examined how elements of urban form, including density, relate to sustainability, especially social sustainability aspects like social equity (access to services and facilities), environmental equity (green and open spaces), and community stability (perceptions of safety and social interaction). They suggest that high-density neighborhoods may be less supportive of socially sustainable behaviors and attitudes than low-density areas. The study shows how physical and non-physical neighborhood features, such as design, maintenance, and safety, are interconnected. People tend to engage socially in their neighborhoods when they can safely walk to accessible services and facilities. They conclude that while the compact city model offers several sustainability benefits, its impact on social sustainability is not entirely positive. Many questions about its broader applicability and replicability remain in an increasingly globalized and urbanized world. They also highlight that privacy—both at home and in public spaces—is crucial for users to feel safe and comfortable [38].

Silvestre and Țîrcă (2019) [39] reviewed various innovations for sustainable development (SD) in existing literature. They emphasize the need for further research to explore how different types of innovations, such as traditional, green, social, and sustainable—and their focus areas—like technology, management, and policy—affect the four TCOS innovation uncertainties: technological, commercial, organizational, and societal. Their insights offer practical guidance on reducing or eliminating these uncertainties by considering an innovation's specific type and focus for SD. The literature they examined addresses urgent environmental and social challenges, and its findings, recommendations, and contributions foster progress toward a sustainable society through innovation and change. This enhances understanding of the most relevant innovations, supporting research and practice in addressing critical sustainability issues faced by society [39].

Akbar and others (2019) [40] created historical land use and land cover (LULC) maps for 1988–2016 and projected future LULC up to 2040 using remote sensing and a Markov model. They investigated how changes in LULC affected urbanization and the economy in Lahore, Pakistan. The study revealed significant land cover modifications, with many areas turning into built-up urban land, which improved residents' living standards and

influenced the real estate market. Barren lands were developed into housing, commercial, and industrial zones, fueling urbanization, industrial growth, and population migration. These changes increased pressure on natural resources and created new challenges for sustainable development. The literature indicates a positive association between urbanization, building growth, and economic development through increased commercial spaces [40].

Eriksson et al. (2021) [41] emphasize that sociodemographic factors influence neighborhood social capital. Policies should promote diverse sociodemographic communities to strengthen social capital, fostering sustainable social development. Encouraging community involvement through the creation of gathering spaces and improved access to shops, cafes, and restaurants can boost social interactions and overall neighborhood satisfaction [41]. Stessens, Canters, and Khan (2021) argue that green spaces benefit human well-being, but non-affluent neighborhoods often lack sufficient access to these areas [42]. Guinaudeau and others (2023) and Inançoglu and colleagues (2023) highlight that urban green and blue spaces offer health benefits and are vital for maintaining cities' quality of life and sustainability [43,44].

Chiaradia and others (2024) [45] indicate that neighborhoods with strong cultural and public services, accessible outdoor spaces, and small retail shops are more inviting for walking and cycling. Their walkability and variety of amenities encourage lively street scenes and regular social interactions, allowing residents to enjoy their preferred lifestyles. Usually, these neighborhoods attract wealthier, better-educated residents who can afford higher housing costs to access amenities, cultural venues, shops, and infrastructure [45]. Dai and others (2024) demonstrate that high-quality neighborhoods can enhance social interaction [46]. Al-Zghoul and Al-Homoud (2025) highlight that walkable environments enhance mobility and play a vital role in promoting social cohesion and interaction [47]. Orlandi and others (2025) argue that community engagement and resilience in public spaces show that users feel safe and secure [48]. Dunuwila and others (2025) [49] state that positive social impacts highlight opportunities to enhance human well-being and provide a broad view of a product's overall social impact. Although the literature indicates that standards for defining positive impacts and methods for assessing them are still evolving and have gaps [49].

Despite extensive studies on sustainable neighborhoods that various NSATs can evaluate, a gap remains in understanding and examining their social and economic impacts on surrounding ar'eas. Furthermore, there is limited research on sustainable neighborhoods' social and economic impacts internationally and regionally. Although there are many certified projects worldwide, and in Oman, there are two certified neighborhoods in Muscat. Therefore, this study addresses these gaps by analyzing and identifying certified sustainable neighborhoods' social and economic impacts on the surrounding area. This will assist urban planners and decision-makers in making better choices that benefit society and promote sustainable development.

## 3. Methodology

This study investigates the social and economic impacts of certified sustainable neighborhoods, specifically "Al-Mouj Muscat," on the nearby urban area "Al-Mawaleh North." It highlights key factors and how sustainable practices influence economic growth, social dynamics, and community well-being. A research question has been formulated: How do sustainable neighborhoods impact the social and economic development of surrounding urban areas? Several objectives are outlined to address this, including analyzing land use and land cover in the Al-Mawaleh North district—the nearest urban area to the sustainable neighborhood Al-Mouj—over 15 years; evaluating the social and economic impacts on the nearby urban area of the certified sustainable neighborhood Al-Mouj; exploring residents'

perceptions and opinions about Al-Mouj's impact on quality of life, community engagement, and social networks; and investigating changes in property values, local business performance, and the growth of surrounding businesses of Al-Mouj. A mixed-methods approach combining quantitative and qualitative methods has been employed to achieve the research aim and objectives and address the research question. This approach enables understanding local conditions and variation, leading to more reliable urban planning interventions [47].

The research employed four data collection methods: literature review to examine whether there are specific criteria for assessing the social and economic impacts of urban development, whether sustainable or not; a case study for spatial analysis to investigate land use and land cover change over the past 15 years to determine the impact of sustainable development on the surrounding area; a questionnaire to explore residents' perceptions and opinions about Al-Mouj's social and economic impacts; and interviews with experts to investigate economic changes and the impact of Al-Mouj on surrounding businesses. The questionnaire resulted in 511 responses, and the interview had 4 responses. To ensure that the sample size of the questionnaire represents the studied population, Cochran's formula is used to determine the sample size needed for a large population with a 95% confidence level and a 5% margin of error. It shows below that a sample size of 384 responses is sufficient, but we received more responses, totaling 511.

$$Samplesize = \frac{z^2 \times P(1-P)}{e^2}$$

Z = z-value for a confidence level of 95% = 1.96
P = Population Proportion (because it is unknown) = 0.5
e = Margin of error = 0.05

$$Samplesize = \frac{1.96^2 \times 0.5(1-0.5)}{0.05^2}$$

$$Samplesize = 384.16$$

The collected data were analyzed to achieve the research objectives. Land use and land cover were analyzed geographically for 2009, 2014, 2019, and 2024 using ArcGIS Pro V3.4 to identify spatial changes over 15 years. Then, the data collected from the questionnaire were statistically analyzed with SPSS to ensure the reliability of the collected data, find out the correlation, and measure the strength and direction of the linear relationship between questions 7 to 12 that measure the social impact. Furthermore, the collected data from open-ended questions in the questionnaire and the interview were also analyzed thematically with NVivo V14 to categorize the responses into specific themes accurately, regarding the large sample size, to achieve reliable findings regarding the social and economic impact.

The literature review shows that no specific criteria have been tested to measure sustainable development's social and economic impact accurately. On the other hand, it indicates a consensus that social impact can be assessed by examining sustainable features, inclusive communities, social cohesion, quality of life, well-being, and mental and physical health. Economic impact can be measured by analyzing effects on the local economy, job opportunities, investments, land use development, and the real estate market. Therefore, these are the chosen criteria to be tested in this study through land use and land cover analysis, the questionnaire, and interviews.

This study employed ethnographic methods to maximize data collection from open-ended questions in the questionnaire and interview. These methods helped identify the research problem, formulate questions for the questionnaire and interview, collect data,

write memos with initial coding into categories, develop detailed memos, refine categories, create diagrams of concepts, and document the findings [50]. They also facilitated collecting participants' observations, reflecting on them, and developing insights into the social and economic impacts of the sustainable neighborhood on the surrounding area. Additionally, this approach allowed the study to provide researchers and stakeholders with reliable indicators to evaluate this impact in other case studies and guided their decision-making.

## 4. Case Study

The selected case study is Al-Mawaleh North district in Wilayat Al Seeb, Muscat Governorate, Oman, as shown in Figure 1a. It was chosen because it is a nearby district within 3 km of Al-Mouj, as shown in Figure 1b, to investigate the social and economic impact of sustainable neighborhoods on the surrounding urban area. Al-Mouj Muscat is recognized as Oman's first certified sustainable neighborhood. It received two BREEAM Communities certificates for the master planning of a larger community of buildings, with a Pass and a Very Good rating for Golf Beach Residences—a collection of luxury family homes on the oceanfront [51]. Golf Beach Residences includes 51 apartments and villas within private gated communities. While designs vary, all are built to the highest standards using premium materials, focusing on long-term energy efficiency, health, and well-being. Residents' living environments help protect the broader environment by reducing energy consumption and $CO_2$ emissions. Al-Mouj also lowers owners' operating costs through smart energy management systems and innovative architectural design. Alongside the ocean and 6 km of beaches, there are tree-lined walkways, nine landscaped parks with eight kids' play areas, and 1.5 km of cycle paths—providing opportunities to enjoy the outdoors and maintain a healthy lifestyle. Al-Mouj includes 8000 residential properties and over 19,000 residents from 85 different countries. It also attracts visitors to its boulevard-style, world-class retail, dining, and entertainment venues, including a 400-berth marina and an award-winning golf course [52].

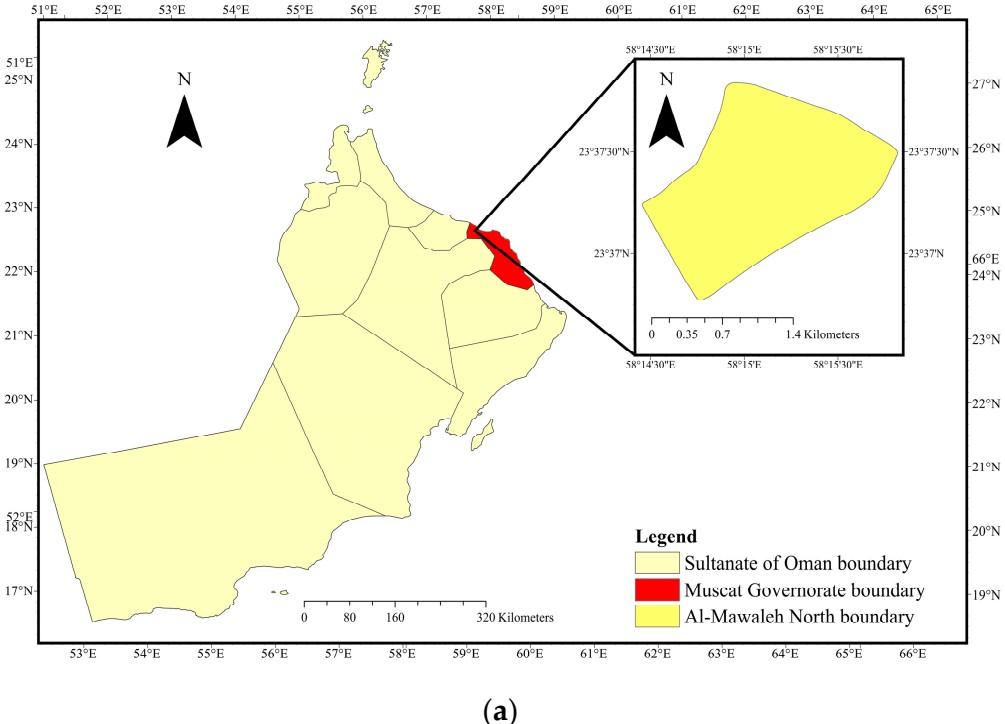

(a)

**Figure 1.** *Cont.*

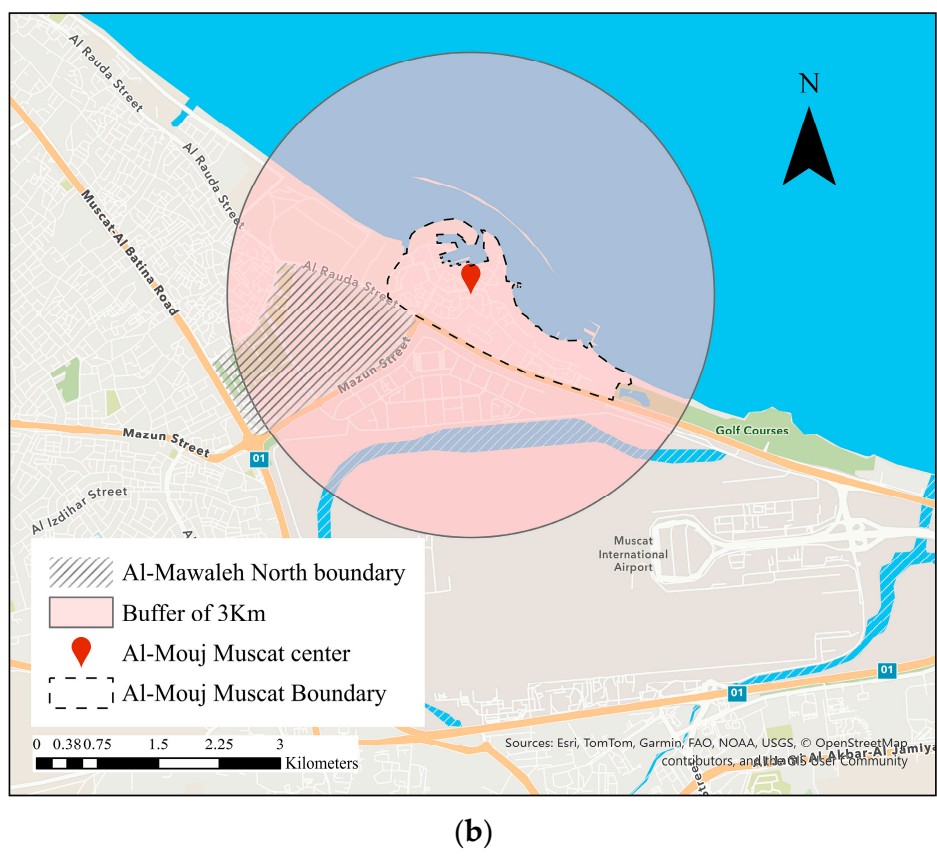

**(b)**

**Figure 1.** Shows the location of Al-Mawaleh North: (**a**) within the Muscat governorate of Oman; (**b**) within 3 Km from Al-Mouj. Source: Authors' elaboration based on primary geographic data and coordinates with permission from GIS Department at Muscat Municipality.

Al-Mouj is Oman's first Integrated Tourism Complex (ITC) project, which began construction in 2007 and was 75% complete by 2020 [53]. Aiming to promote economic diversification, encourage tourism growth and employment, and create opportunities for enterprise and investment, in line with Oman Vision 2040. Therefore, the project is designed to be Oman's leading lifestyle and leisure destination, attracting people from different backgrounds, cultures, and industries to connect and thrive within a balanced community. Additionally, this community plans to integrate facilities and services seamlessly to ensure a sustainable and livable environment. The Royal Decree issued in 2006 permits foreigners to own real estate within Integrated Tourism Complexes (ITCs) like Al-Mouj, granting them freehold ownership rights. They can apply for long-term residency for themselves and their immediate family if they continue to own property [54].

Al-Mouj was chosen as a case for this study; however, as previously mentioned, there are two certified neighborhoods in Muscat. This is because it is designed as a private gated community within the residential area to ensure residents' privacy. It also functions as an open community with public and recreational spaces for visitors. This setup allows the research to investigate the social and economic impacts on the surrounding area. In contrast, the other sustainable neighborhood in Muscat, Ras Al Hamra (RAH) development project, is a private, enclosed community exclusively for Petroleum Development Oman (PDO) employees and their families, which prevents investigating the social and economic impacts on the surrounding area [55]. Both neighborhoods are certified as sustainable developments, but this study aims to assess the impact beyond the boundary to maximize the benefits of sustainability, which is far from selfish.

The inclusive development and design of Al-Mouj aim to meet the diverse needs of residents and visitors, facilitating the examination of social and economic impacts on

the surrounding area. Additionally, it is the first ITC that allows foreigners to own real estate, making it not only a sustainable development but also an inclusive community, with evidence of success over the past 15 years that can support the achievement of the research objectives. There is a limitation in having only two cases to choose from. However, focusing the study on one case and investigating it from different perspectives enables obtaining reliable results that provide planners and decision-makers with valuable insights into future decisions. In the next part, we started the investigation by analyzing the land use and land cover of the surrounding area, Al-Mawaleh North, using ArcGIS Pro V3.4 to highlight the impact on land development.

## 5. Land Use and Land Cover

A spatial analysis of land use and land cover change in Al-Mawaleh North is conducted to achieve the research objectives. Four years were selected to illustrate changes and development over 15 years in Al-Mawaleh North, aiming to record the impact of Al-Mouj on surrounding land use development. The selected years are 2009, 2014, 2019, and 2024, as shown in Figure 2a–d, each spaced five years apart to observe significant land use and cover changes over consistent time intervals. Primary geographical data and coordinates for these years were obtained from a reliable government agency, Muscat Municipality. Land use codes were assigned based on land type using Python V3.14 to create a new attribute field. Land use maps were then generated with ArcGIS. These spatial analysis maps clearly display land use changes and development, including the increase in buildings over 15 years, classified into residential (low and medium density), commercial, mixed-use, educational, public buildings, healthcare, social areas, and religious use. Table 1 presents changes in land use type across the four selected years.

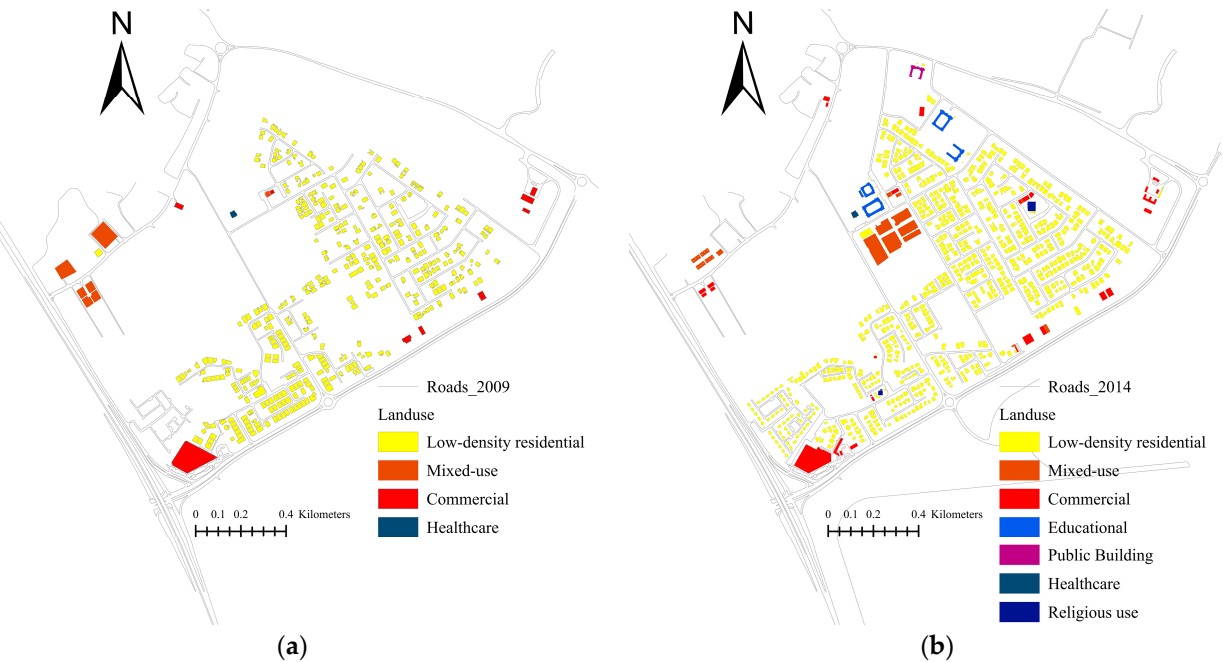

**(a)**   **(b)**

**Figure 2.** *Cont.*

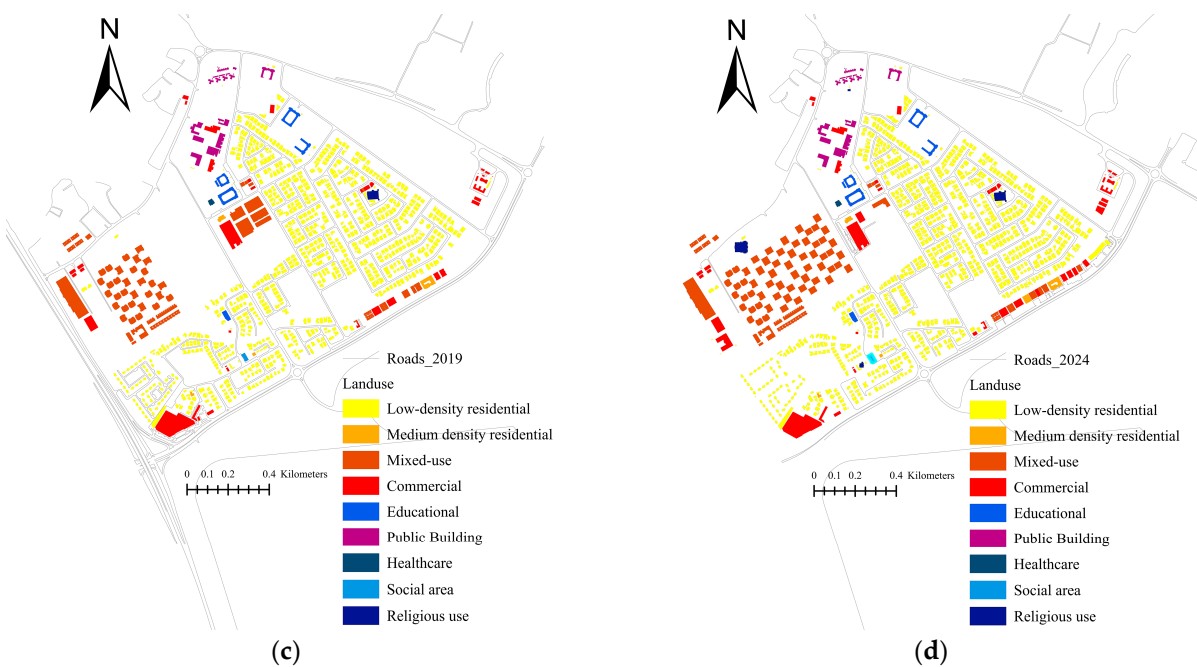

**Figure 2.** Presents the land use in Al-Mawaleh North: (**a**) 2009; (**b**) 2014; (**c**) 2019; (**d**) 2024. Source: Authors' elaboration based on primary geographic data and coordinates with permission from GIS Department at Muscat Municipality.

**Table 1.** Shows the changes in the number of units for each use in 2009, 2014, 2019, and 2024. Source: Authors' elaboration based on primary geographic data and coordinates with permission from GIS Department at Muscat Municipality.

| Land Use/Year | 2009 | % Change from 2009 to 2014 | 2014 | % Change from 2014 to 2019 | 2019 | % Change from 2019 to 2024 | 2024 |
|---|---|---|---|---|---|---|---|
| Residential | 556 units | 91.18% | 1063 units | 13.73% | 1209 units | 5% | 1269 units |
| Commercial | 9 units | 288.89% | 35 units | 31.42% | 46 units | 13.04% | 52 units |
| Mixed-use | 8 units | 0% | 8 units | 112.5% | 17 units | 0% | 17 units |
| Educational | 0 unit | 400% | 4 units | 25% | 5 units | 0% | 5 units |
| Public Buildings | 0 unit | 100% | 1 unit | 300% | 4 units | 0% | 4 units |
| Healthcare | 1 unit | 0% | 1 unit | 0% | 1 unit | 0% | 1 unit |
| Religious Building | 0 unit | 200% | 2 units | 50% | 3 unit | 33.33% | 4 units |
| Social Area | 0 unit | 0% | 0 unit | 100% | 1 unit | 0% | 1 unit |
| Total number | 574 | 94.07% | 1114 | 15.44% | 1286 | 5.21% | 1353 |

The land use and land cover maps analysis shows that residential land use experienced the most significant change and growth, increasing from 556 units in 2009 to 1269 in 2024, including single villas, twin villas, and apartments. There has also been a rise in the area's population density as shown in Table 1. Meanwhile, commercial land use ranked second in building expansion, growing from 9 units in 2009 to 52 in 2024, including shops, companies, and offices. Mixed-use buildings with commercial and residential uses ranked third, rising from 8 units in 2009 to 17 in 2024. Additionally, the number of educational campuses—comprising private schools, government schools, and colleges—grew from zero units in 2009 to five in 2024. Public and religious buildings also increased from zero units to four over the fifteen years. While many land uses developed over time, some remained unchanged, like healthcare, which maintained only one building throughout the period. Conversely, the social area for indoor meetings and social activities expanded by one building in 2019. The total number of buildings increased by 135.7% between 2009 and 2024.

The first year selected was 2009, two years after Al-Mouj's construction began, as shown in Figure 2a. The area was primarily residential, with one mall on the corner, some limited shops, and mixed-use spaces. By 2014, as shown in Figure 2b, significant changes occurred, with residential areas increasing by 91.18%, commercial areas by 288.89%, educational spaces by 400%, and public buildings by 100%. By 2019, as shown in Figure 2c, the number of mixed-use buildings grew by 112.5%, with developments adopting a style closer to the Al-Mouj community. By 2024, as depicted in Figure 2d, commercial activity continued to expand, including shops and restaurants along the main street leading to the Al-Mouj entrance, with nearly all unused spaces now fully occupied. The four maps demonstrate the accuracy of the data record; however, some mixed-use spaces appear to have disappeared from 2009 to 2014. After investigation, it was found that this was a temporary use removed by the Muscat Municipality. There were some limitations regarding data collection, since the only source is the Muscat Municipality, which has maintained these records over the years. In the next section, the questionnaire will explore participants' perspectives on the social impacts of Al-Mouj and whether they perceive an economic impact based on the recorded growth of commercial activities from land use and land cover analysis.

## 6. Questionnaire

The questionnaire was created to meet research objectives by collecting participants' perceptions of Al-Mouj's social impact on the nearby area. A QR code linking to the questionnaire was provided in Al-Mouj's public spaces for over six months. It was fixed in the designated area for public advertising in Al-Mouj within the public and commercial spaces to convey the required intention. Furthermore, it had been distributed in Al-Mawaleh North to be fixed in the commercial centers, shops, and restaurants, to make sure it would be visible for the residents and visitors, and they can interact with it without any intervention from the research team, to make sure that there is no bias in the collected sample.

The questionnaire was a blind investigation; the research team could not identify the participants from the questions. Participants were 18 years or older, including both men and women. The survey included 14 open-ended and closed-ended questions, written in Arabic and English to ensure clarity for all respondents. The questions covered age and gender to identify the main group that frequently interacts with Al-Mouj's public and commercial spaces, and nationality to determine whether Al-Mouj's inclusive community is designed for locals or foreigners. Additionally, they asked whether participants live near Al-Mouj or far away to assess how well Al-Mouj serves the surrounding area, and how often they visit to determine the level of interaction, whether frequent or rare. To meet the research objectives of exploring the social impact, the questions addressed participants' perspectives on quality of life in Al-Mouj, community engagement, and social networks, and how these compare to other places in Oman. The survey also inquired about features of Al-Mouj that influence their experience during visits, whether they would like to live in a similar community, and if urban design features, walking paths, and green spaces positively affect mental and physical health. Furthermore, it asked if public spaces at Al-Mouj help reduce social isolation and foster social connections, and whether the facilities and design support a diverse and inclusive community with various needs and backgrounds, including individuals with disabilities.

To address the economic impact, the questionnaire asked if they support projects like Al-Mouj in Oman, have observed increased economic activity, or believe Al-Mouj has created more job opportunities for the local community. Most questions were closed-ended to ensure reliable, accurate, and precise results, except for one that asked participants to explain why they would or would not like to live in Al-Mouj, allowing them to share their perceptions of the community. The closed-ended questions were easy for participants to

answer, did not require prior experience, and aligned with the research objectives. The open-ended questions were reserved for interviews with specialist-selected participants.

The analysis of responses which is shown in Appendix A, Tables A1 and A2 and presented in Figure 3a,b shows that 52.1% of participants are female, and 67.3% are Omani. 87.9% live near Al-Mouj. The largest age group is 35 to 44, making up 41.3%. Regarding frequent visits to Al-Mouj, 35% go 4–6 times weekly. Participants' favorite features at Al-Mouj include walkability and cycling paths, parks and recreational areas, and community centers for events and activities. 97.1% believe they experience a better quality of life in Al-Mouj compared to other places in Oman. 83.2% would like to live in a community like Al-Mouj. 56.9% think that the public spaces at Al-Mouj help reduce social isolation and foster social connections. 69.7% feel that the facilities and design support a diverse and inclusive community with various needs and backgrounds, including those with disabilities. 65.4% believe that urban design features, walking paths, and green spaces positively impact mental and physical health. 98.4% want to see more projects like Al-Mouj in Oman. 84.5% noticed increased economic activity in the area. 63.8% believe that Al-Mouj has created more job opportunities for the local community.

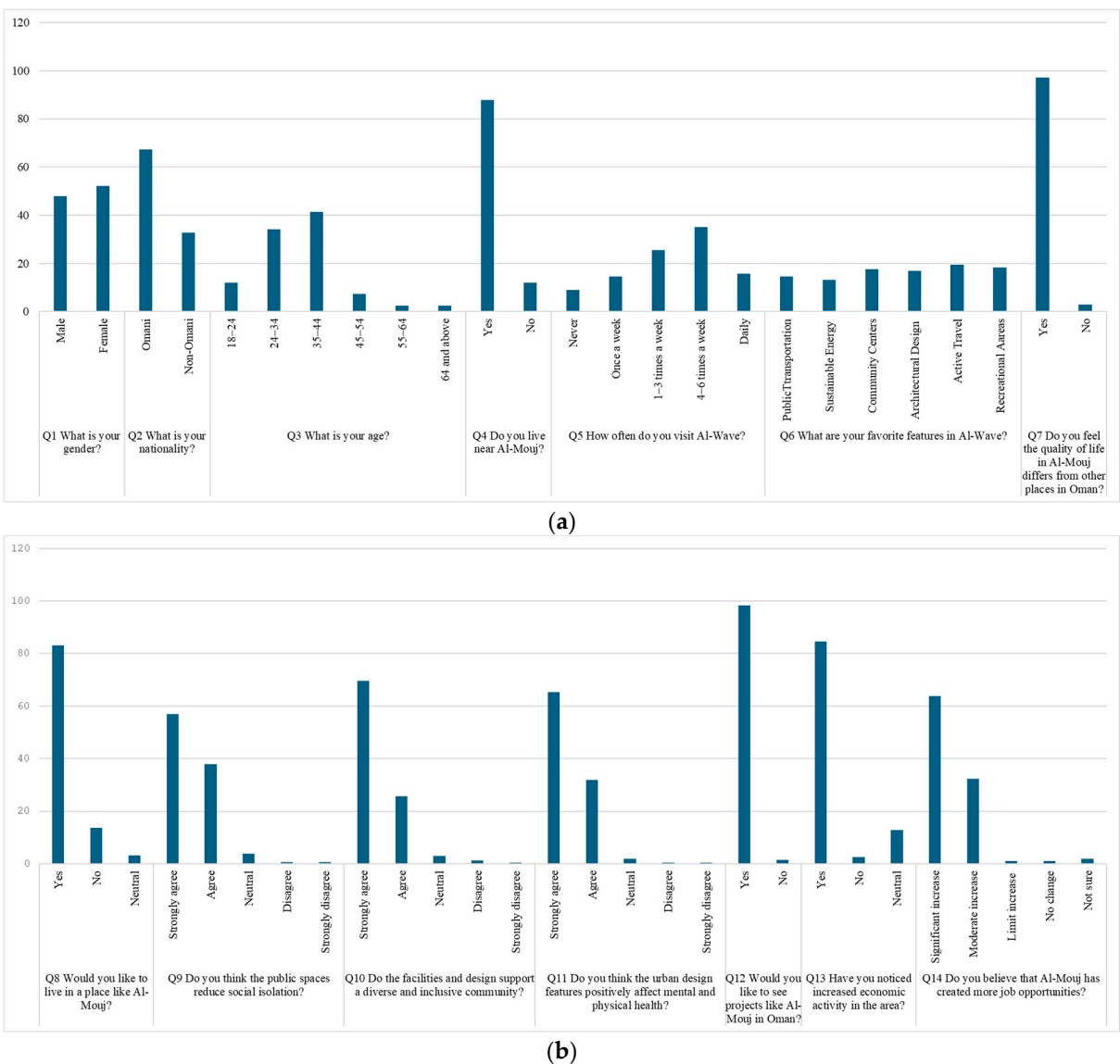

**Figure 3.** Displays responses to the questionnaire: (**a**) questions 1 through 7; (**b**) questions 8 through 14. Source: Authors' elaboration.

The questions differed in terms of answers. Three of the questions, 9, 10, and 11, which assessed quality of life and explored perceptions regarding public spaces that reduce social isolation, facilities and designs that foster a more diverse and inclusive community, and features like walking paths and green areas that positively affect mental and physical health, used the 5-point Likert scale. Cronbach's Alpha was calculated using IBM SPSS Statistics V30 to evaluate the internal consistency and reliability of the 5-point Likert scale for these questions to ensure that the research objective of investigating the social impact has been achieved and the results are reliable. As shown in Table 2 the resulting Cronbach's Alpha (0.911) indicates excellent internal consistency and reliability for the scale.

**Table 2.** Reliability Statistics for the 5-point Likert Scales of Questions 9, 10, and 11. Source: Authors' elaboration generated by SPSS.

| Cronbach's Alpha | Cronbach's Alpha Based on Standardized Items | N of Items |
|---|---|---|
| 0.906 | 0.911 | 3 |

Correlation analysis measured the strength and direction of the linear relationship between questions from 7 to 12. Table 3 shows the results, which include three types of correlations: strong positive, positive, and negative. The strong positive correlation indicates that responses supporting the idea that Al-Mouj's urban design features promote mental and physical health are also linked to increased opportunities for fostering a diverse and inclusive community. The negative correlation reveals that respondents who do not like living in a community like Al-Mouj agree that its facilities and design support a diverse and inclusive community.

**Table 3.** Presents the correlation between questions from 7 to 12. Source: Authors' elaboration generated by SPSS.

| | | Q7 Do You Feel the Quality of Life in Al-mouj Differs from Other Places in Oman? | Q8 Would You Like to Live in a Place Like Al-Mouj? | Q9 Do You Think Public Spaces Reduce Social Isolation? | Q10 Do the Facilities and Design Support a Diverse and Inclusive Community? | Q11 Do You Think Urban Design Features Positively Affect Mental and Physical Health? | Q12 Would You Like to See Projects Like Al-Mouj in Oman? |
|---|---|---|---|---|---|---|---|
| Q7: Do you feel the quality of life in Al-Mouj differs from other places in Oman? | Pearson Correlation | 1 | 0.148 ** | 0.084 | 0.064 | 0.075 | 0.445 ** |
| | Sig. (2-tailed) | | <0.001 | 0.059 | 0.150 | 0.092 | <0.001 |
| Q8: Would you like to live in a place like Al-Mouj? | Pearson Correlation | 0.148 ** | 1 | 0.019 | −0.049 | 0.023 | 0.214 ** |
| | Sig. (2-tailed) | <0.001 | | 0.673 | 0.274 | 0.597 | <0.001 |
| Q9: Do you think public spaces reduce social isolation? | Pearson Correlation | 0.084 | 0.019 | 1 | 0.696 ** | 0.709 ** | 0.051 |
| | Sig. (2-tailed) | 0.059 | 0.673 | | <0.001 | <0.001 | 0.246 |
| Q10: Do the facilities and design support a diverse and inclusive community? | Pearson Correlation | 0.064 | −0.049 | 0.696 ** | 1 | 0.913 ** | 0.073 |
| | Sig. (2-tailed) | 0.150 | 0.274 | <0.001 | | <0.001 | 0.101 |
| Q11: Do you think urban design features positively affect mental and physical health? | Pearson Correlation | 0.075 | 0.023 | 0.709 ** | 0.913 ** | 1 | 0.083 |
| | Sig. (2-tailed) | 0.092 | 0.597 | <0.001 | <0.001 | | 0.062 |
| Q12: Would you like to see projects like Al-Mouj in Oman? | Pearson Correlation | 0.445 ** | 0.214 ** | 0.051 | 0.073 | 0.083 | 1 |
| | Sig. (2-tailed) | <0.001 | <0.001 | 0.246 | 0.101 | 0.062 | |
| N | | 511 | 511 | 511 | 511 | 511 | 511 |

** Correlation is significant at the 0.01 level (2-tailed). Green, strong positive correlation; yellow, positive correlation; red, negative correlation.

Thematic analysis using NVivo V14 software was conducted on 511 responses to the open-ended question about whether participants would like to live in a place like Al-Mouj,

along with their reasons for or against it. The responses were collected and combined into a single file. This file was imported into NVivo to code and analyze similar responses under main categories. The analysis showed that respondents favored living in Al-Mouj due to the quality of life, mixed land use, architectural designs, active transportation infrastructure, and green and public spaces. Conversely, reasons against included cultural barriers related to differing lifestyles from Omani culture, traffic congestion caused by the roundabout before the entrance of Al-Mouj, lack of privacy in some areas, and the high cost of rent or ownership. Neutral responses were also attributed to cultural barriers, privacy concerns, and unaffordability. Figure 4a,b present the questionnaire's thematic analysis results.

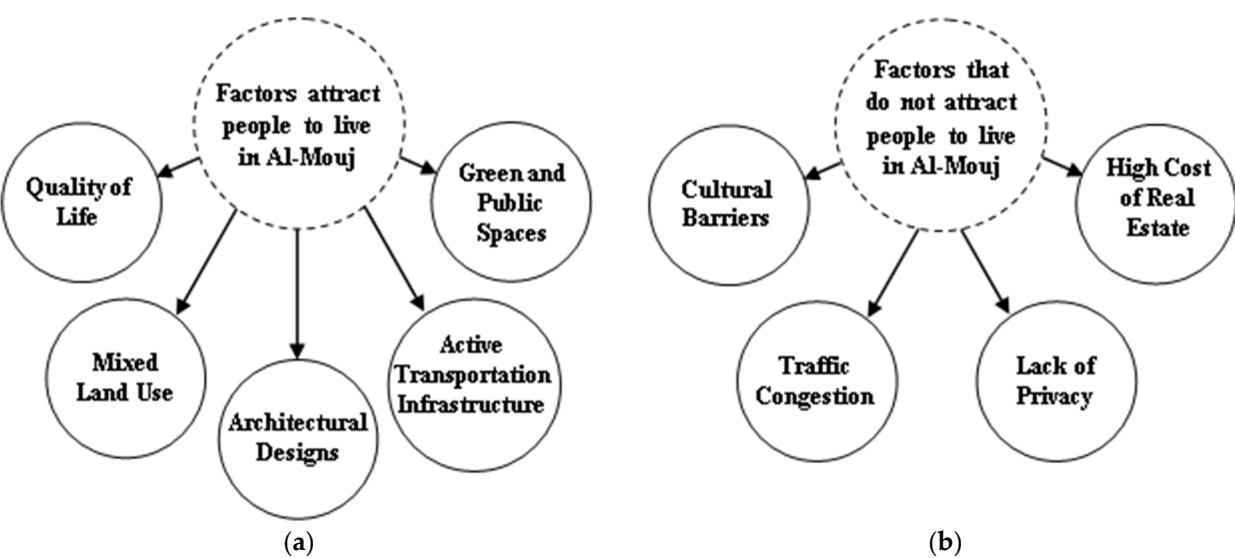

(a)  (b)

**Figure 4.** (**a**) Shows factors that attract people to live in Al-Mouj. (**b**) Shows factors that discourage people from living in Al-Mouj. Source: Authors' elaboration.

## 7. Interview

A structured interview was conducted to investigate the economic impact of Al-Mouj on local businesses and the surrounding real estate. The interview included seven questions about the interviewee's occupation, how long they have worked in the area, whether they noticed any specific changes or developments in their business since the establishment of the Al-Mouj project, how Al-Mouj has affected economic activities in the area, their views on competition among entrepreneurs and business owners in the area, their perspectives on the real estate market and rental prices near Al-Mouj compared to other parts of Muscat, and whether they believe sustainable projects like Al-Mouj attract foreign investments and appeal to investors and users. Participants were chosen because they had been business owners in Al-Mouj or nearby for at least 12 years, ensuring they had observed economic growth since the start of the Al-Mouj project or earlier. Four business owners specializing in real estate, construction, and engineering consultancy were interviewed. These four interviews provided valuable qualitative insights from industry experts regarding Al-Mouj's impact on local economic activity.

A thematic analysis of the qualitative data collected from the interviews was conducted. The responses were transcribed, reviewed, and combined into a single file for comparison. Six main themes were identified: economic impact, real estate market impact, sustainability as a driver, increased market competition, and urban development. Figure 5 displays the six themes derived from the interview analysis, including quotes from interviewees.

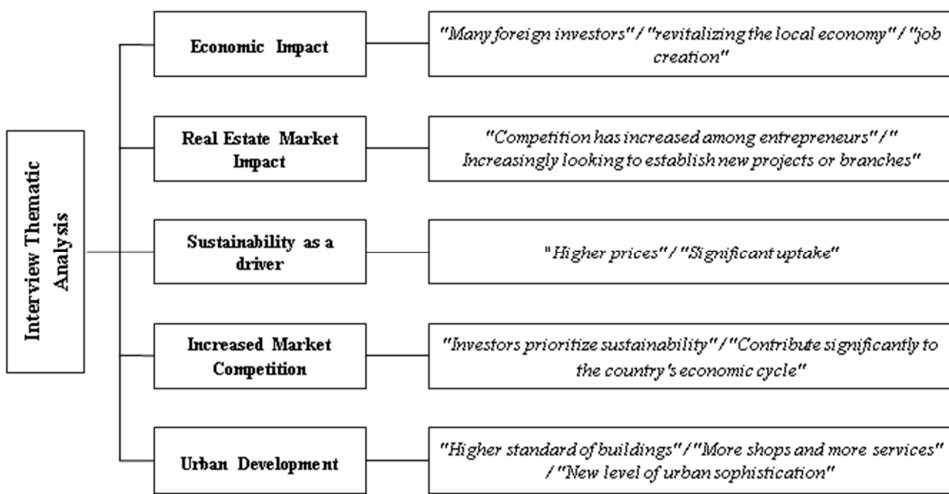

**Figure 5.** Shows the six themes identified from the interview analysis and quotes from interviewees. Source: Authors' elaboration.

## 8. Results and Discussion

The spatial analysis of the case shows that from 2009 to 2024, the Al-Mawaleh North area experienced significant growth in the number of buildings and land use diversity, starting with the development of Al-Mouj. The expansion of residential areas indicates that the region has attracted many residents, leading to population growth and increased demand for services and land uses supporting infrastructure, such as improved educational, public, healthcare, and religious facilities. This expansion and rise in demand drive continued growth and a rise in commercial activities, including mixed-use developments, companies, offices, restaurants, and shops, which reflect economic growth in Al-Mawaleh North and demonstrate a diverse, investment-friendly local economy. Furthermore, it supported the commercial activities of Al-Mouj and attracted local, regional, and international investments, fostering growth in the local market and creating new job opportunities. The sustainable development of Al-Mouj as a mixed-use area with various-sized spaces for economic activities and a welcoming community enabled it to attract different levels of investment. Moreover, the high demand for more economic activities at affordable prices from residents of Al-Mouj and Al-Mawaleh allowed the two communities to exchange benefits, thereby maximizing the impact of sustainable economic growth, extending advantages to nearby neighborhoods, and supporting local economic development.

The questionnaire showed that Al-Mouj's public spaces and facilities effectively serve local needs, with 67.3% of participants being locals. The spatial analysis also indicates that the development of Al-Mawaleh North has left no green, open, or public spaces for recreation. Conversely, the questionnaire revealed that Al-Mouj's spaces and amenities are accessible to people beyond its residents, as 87.9% of participants are visitors from nearby areas, mainly from Al-Mawaleh North—where the questionnaire QR was distributed—who come to Al-Mouj for social and economic activities during the week, mostly 4 to 6 times per week, and on weekends when special events are held for different age groups. Furthermore, most participants believe that living in Al-Mouj or nearby offers a higher quality of life and show interest in living in similar communities, trusting that public spaces and facilities can boost visitors' mental and physical health, not just those of residents. Additionally, the responses highlight a strong connection between quality of life and its impact on social interaction, health, and economic development, with 84.5% noting increased economic activity and employment opportunities. This illustrates that the social benefits of Al-Mouj's sustainable development extend to the surrounding area and make up for its shortage of public spaces and facilities.

The correlation and thematic analysis of the questionnaire show that, although locals enjoy and benefit from Al-Mouj's public spaces and facilities, they still perceive cultural barriers to living there. The living spaces were designed based on international standards, which differ from traditional preferences. Privacy in Al-Mouj varies from what locals prefer, especially regarding residential spaces and building designs. This results from developing Al-Mouj mainly according to international sustainable standards that differ from local ones. However, BREEAM standards [29] consider local aspects; they do not fully meet local standards. This may also be because Al-Mouj was developed as a sustainable ITC project that attracts people from different backgrounds, cultures, and industries to promote economic diversification and create opportunities for enterprises and investment. Although Al-Mouj cannot fully meet local needs, it has attracted residents from 85 different countries, supported the local economy, and positively impacted the surrounding area.

The thematic analysis of the questionnaire also shows that traffic congestion on the main street outside Al-Mouj—which is the only link between Al-Mouj and Al-Mawaleh North, and the area with the most commercial activity in Al-Mawaleh—has increased due to growing population, rising demand, and the strong, dependent relationship between the two communities for social and economic benefits, leading to traffic issues. During rush hours, the current infrastructure becomes overwhelmed, especially at peak times, and a solution from the municipality is needed to improve the connection between Al-Mouj and Al-Mawaleh North. This also clearly demonstrates the dependent relationship between sustainable development and the surrounding area, as when designed as open communities, they become attractive points and impact the surrounding area.

Real estate prices within Al-Mouj are not affordable, which has prompted many development projects to be built around Al-Mouj at more reasonable prices. These projects attract high demand from locals who want to live in a traditional and local style, while also benefiting from social facilities and economic activities near or within sustainable development. Additionally, the economic activities within Al-Mouj attract international brands mainly because of the high rent or ownership costs, encouraging local brands to open outside Al-Mouj, close enough to serve residents with more competitive prices. This has contributed to the growth of local brands in the surrounding area.

The interviewees emphasized Al-Mouj's economic influence based on their experience and recent observations of the nearby area. Participants agreed that the area has experienced significant economic growth over the past 12 years. These changes include various factors, such as increased economic activities, higher foreign and local investments, and improved employment opportunities. The presence of Al-Mouj has helped boost competitiveness among entrepreneurs and investors to start new projects and branches of well-known international restaurants and stores on Al-Mouj Street, located in Al-Mawaleh North. It has also attracted many young people to start small ventures across from Al-Mouj, bringing in residents and encouraging young entrepreneurs to develop and launch new investment projects as demand increases. The interviewees have observed a rise in property values around Al-Mouj compared to other parts of Muscat Governorate. This is due to the high standards of buildings that closely match Al-Mouj's criteria, attracting locals who want to live at a similar sustainable level but with a local style, along with the proximity to many services offered by Al-Mouj and its surrounding area. This demonstrates Al-Mouj's substantial economic impact on the nearby region.

## 9. Conclusions

The results and discussion show that Al-Mouj introduced a new lifestyle through mixed-use development to the local community, supporting the economy and fostering social cohesion. It has also shifted local consumer preferences toward sustainable practices that

impact both the short and long term, offering a new perspective on quality of life. Although this new perspective may differ from traditional standards, it remains somewhat acceptable to the local society. It can still improve residents' and visitors' mental and physical health in the surrounding area. Al-Mouj's sustainable development has attracted locals and people from different countries to its unique lifestyle. While international assessment tools like BREEAM consider Al-Mouj a private gated community, its privacy remains negotiable for the local community. Privacy varies in its forms and meanings across different societies and cultures. This emphasizes the need for local standards in Oman and a neighborhood sustainability assessment tool that considers local needs and encourages sustainable practices, enabling more projects to meet the specific needs of the Oman community.

This study established reliable criteria for assessing a sustainable neighborhood's social and economic impacts on its surrounding area. It demonstrates that sustainable development can boost local economic growth by creating opportunities for healthy competition among international and local entrepreneurs and increasing job opportunities. It also supports economic sustainability by strengthening the local economy for current residents and future generations through sustainable solutions and infrastructure. While Al-Mouj has become a sustainable community that supports the local economy, it has not been able to provide affordable housing. It was intended to be carefully managed so everyone could afford to live there. Sustainable practices should promote inclusive communities, social cohesion, quality of life, well-being, and mental and physical health beyond certified sustainable neighborhoods and affordable housing within their boundaries.

Oman is developing two new sustainable neighborhoods, Sultan Haitham City and Madinat al Irfan. This study provides decision-makers and urban planners working on these projects and others with reliable criteria tested to maximize sustainable development's social and economic benefits for the surrounding area. These criteria include creating sustainable neighborhoods that are open communities, attracting visitors, users, and regional and global investors, thereby boosting the local economy. Developing sustainable neighborhoods that promote healthy lifestyles is not limited to residents; it also extends to surrounding districts, which can improve social cohesion and enhance the positive social impact of sustainable development. The new sustainable neighborhoods in Oman are located within urban areas that lack spaces for social and recreational activities and are seeking more economic opportunities. Applying the tested criteria from this research will enable these sustainable developments to broaden their influence on the surrounding area and maximize their impact.

Some challenges have arisen during this study in identifying business owners for interviews who have lived in the area for the past 15 years and can serve as witnesses to the economic impact of Al-Mouj on the Al-Mawaleh North district. A future study is needed to compare urban development in Al-Mawaleh North and Al-Mawaleh South within a 7 km radius of Al-Mouj to determine how much the sustainable neighborhood influences the social and economic aspects of the surrounding area. Additionally, future research could explore ways to design local tools for sustainability assessment to establish Omani standards for Neighborhood Sustainability Assessment Tools suitable for the Omani context.

**Author Contributions:** Conceptualization, methodology, validation, analysis, investigation, writing—review and editing, E.H.M.N. and M.A.M.K.; Software, analysis, investigation, data curation, writing—original draft preparation, E.H.M.N. and A.M.A.S.; Supervision, E.H.M.N. All authors have read and agreed to the published version of the manuscript.

**Funding:** The Global College of Engineering and Technology and the Scientific College of Design cover the article processing charge. This research received no external funding.

**Institutional Review Board Statement:** The questionnaire was approved by the Research and Innovation office of Global College of Engineering and Technology (7 January 2025) and the Research & Faculty Promotion Committee of Scientific College of Design (29 January 2025).

**Informed Consent Statement:** Informed consent was obtained from all subjects involved in the interviews.

**Data Availability Statement:** Primary geographical data and coordinates for the study area were obtained from the Muscat Municipality, a reliable government agency. However, these data are not available for open access publication. Further inquiries can be directed to the corresponding author(s).

**Acknowledgments:** The authors would like to acknowledge the Muscat Municipality for providing the primary geographical data and Huda Al-Afifi, the GIS Manager at the Muscat Municipality, for her support in conducting this research.

**Conflicts of Interest:** The authors declare no conflicts of interest.

## Abbreviations

The following abbreviations are used in this manuscript:

| | |
|---|---|
| NSAT | Neighborhood Sustainability Assessment Tools |
| BREEAM | Building Research Establishment Environmental Assessment Method |
| BRE | Building Research Establishment |
| SD | Sustainable Development |
| TCOS | Technological, Commercial, Organizational, and Societal) |
| LULC | Land Use and Land Cover |
| ITC | Integrated Tourism Complex |
| ArcGIS Pro V3.4 | ArcGIS Pro version 3.4 is a comprehensive Geographic Information System software developed by Esri. |
| IBM SPSS V30 | Statistical Product and Service Solutions software version 30. |
| NVivo V14 | Non-numerical Unstructured Data Indexing, Searching, and Theorizing software, version 14. |

## Appendix A

The following tables summarize the questionnaire responses.

**Table A1.** Shows responses to questions 1 through 7. Source: Authors' elaboration.

| Q1 What Is Your Gender? | | Q2 What Is Your Nationality? | | Q3 What Is Your Age? | | | | | | Q4 Do You Live Near Al-Mouj? | | Q5 How Often Do You Visit Al-Wave? | | | | | Q6 What Are Your Favorite Features of Al-Wave? | | | | | | Q7 Do You Feel the Quality of Life in Al-Mouj Differs from Other Places in Oman? | |
|---|---|---|---|---|---|---|---|---|---|---|---|---|---|---|---|---|---|---|---|---|---|---|---|---|
| Male | Female | Omani | Non-Omani | 18–24 | 24–34 | 35–44 | 45–54 | 55–64 | 64 and above | Yes | No | Never | Once a week | 1–3 times a week | 4–6 times a week | Daily | Access to public transportation | Sustainable Energy Usage | Community Centers | Architectural Designs | Walkability and Cycling Paths | Parks and Recreational Areas | Yes | No |
| 47.9% | 52.1% | 67.3% | 32.7% | 11.9% | 34.2% | 41.3% | 7.4% | 2.5% | 2.5% | 87.9% | 12.1% | 9% | 14.5% | 25.5% | 35% | 15.7% | 14.5% | 13.1% | 17.7% | 16.9% | 19.5% | 18.3% | 97.1% | 2.9% |

**Table A2.** Shows responses to questions 8 through 14. Source: Authors' elaboration.

| Q8 Would You Like to Live in a Place Like Al-Mouj? | | | Q9 Do You Think Public Spaces Reduce Social Isolation? | | | | | Q10 Do the Facilities and Design Support a Diverse and Inclusive Community? | | | | | Q11 Do You Think Urban Design Features Positively Affect Mental and PHYSICAL HEALth? | | | | | Q12 Would You Like to See Projects Like Al-Mouj in Oman? | | Q13 Have You Noticed Increased Economic Activity in the Area? | | | Q14 Do You Believe that Al-Mouj has Created More Job Opportunities? | | | | |
|---|---|---|---|---|---|---|---|---|---|---|---|---|---|---|---|---|---|---|---|---|---|---|---|---|---|---|---|
| Yes | No | Neutral | Strongly agree | Agree | Neutral | Disagree | Strongly disagree | Strongly agree | Agree | Neutral | Disagree | Strongly disagree | Strongly agree | Agree | Neutral | Disagree | Strongly disagree | Yes | No | Yes | No | Neutral | Significant increase | Moderate increase | Limit increase | No change | Not sure |
| 83.2% | 13.7% | 3.1% | 56.9% | 38% | 3.9% | 0.6% | 0.6% | 69.7% | 25.8% | 2.9% | 1.2% | 0.4% | 65.4% | 31.9% | 2% | 0.4% | 0.4% | 98.4% | 1.6% | 84.5% | 2.5% | 12.9% | 63.8% | 32.3% | 1% | 1% | 2% |

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
