# Peer review of "Social and Economic Influence of Sustainable Development: The Case of Al-Mouj, Muscat, Oman"

_sustainability, doi:10.3390/su17209037_

Round 1
Reviewer 1 Report
Comments and Suggestions for Authors
In general, this manuscript is clearly written, presents an interesting case study, and aligns well with the scope of the journal. Nevertheless, in its current form it requires further refinement before being suitable for publication in a high-impact venue such as Sustainability. In particular, several sections would benefit from a more in-depth treatment, stronger methodological transparency, and a richer discussion of the implications of the findings. These improvements would significantly increase the scholarly rigor and practical relevance of the paper. To facilitate the revision process, the authors will find attached a PDF file containing detailed, section-by-section comments and suggestions, which are intended to guide them through the necessary adjustments in a constructive and targeted manner. Authors can also find comments below:
1 - I recommend that the authors consider adopting a more engaging title. A clearer and catchier formulation could better capture readers’ attention, enhance the paper’s visibility, and more effectively highlight the study’s main contribution.
2 - I suggest that the authors place greater emphasis on the applications and implications of their research in the final part of the abstract, as this would strengthen the overall impact and relevance of the study.
3 - I recommend that the authors limit the number of keywords to a maximum of five and select them carefully, as these terms play a crucial role in indexing the paper in major academic search engines.
4 - The introduction provides a broad overview of the historical and conceptual foundations of sustainability and urban planning, but at times it reads more like a general textbook summary than a targeted framing of the research problem. While the references to the Brundtland Report, Howard’s “Garden City,” and Perry’s neighborhood unit are useful, the section would benefit from a clearer linkage between these classical theories and the specific context of sustainable neighborhoods in Oman. Furthermore, some parts could be streamlined to avoid redundancy and make space for a sharper articulation of the research gap. Highlighting more recent and internationally relevant literature on neighborhood sustainability would also strengthen the positioning of the study within current academic debates.
5 - In the introduction, where authros state "[...] In recent years, awareness of sustainability issues has increased, as sustainability encompasses various aspects of resource use and society [...]", to better contextualize this sentence in a more recent and internationa literature scenario, the authors shoul cite here the following and very recent paper which addresses the various facets of sustainability in society:
-Falegnami, A., Romano, E. and Tomassi, A., 2024. The emergence of the GreenSCENT competence framework: A constructivist approach: The GreenSCENT theory. In The european green deal in education (pp. 204-216). Routledge. DOI: 10.4324/9781003492597-17
6 - The authors should explicitly state whether the literature review was conducted according to a recognized systematic review protocol (such as PRISMA or ProKnow-C), or whether it should rather be considered a scoping review. Clarifying this aspect would improve the transparency and methodological rigor of the paper. In either case, the authors are encouraged to provide essential details of their search strategy, including the exact query used, the database(s) consulted, and the inclusion and exclusion criteria applied. Such information would not only allow readers to better assess the robustness of the review, but also facilitate reproducibility and increase the credibility of the research process.
7 - The methodology is generally well-structured, combining quantitative and qualitative approaches and employing recognized analytical tools. However, the section would benefit from greater clarity and detail in some areas, for instance, in explaining the rationale for selecting specific data collection methods, and in justifying how the sample size ensures representativeness of the studied population. Moreover, while multiple techniques (GIS, SPSS, NVivo) are mentioned, the link between each method and the corresponding research objective could be articulated more explicitly, thereby reinforcing the methodological rigor and transparency of the study.
8 - The case study paragraph is clearly described and provides useful contextual information about Al-Mouj and its surrounding area. Nonetheless, the section could be strengthened by clarifying the criteria that guided the choice of this specific case, and by better linking its characteristics to the research objectives. A more critical reflection on the representativeness and possible limitations of focusing on a single case would also enhance the robustness of the study.
9 - Figure 1 is of rather poor quality, and I kindly ask the authors to provide a higher-resolution version. In addition, they might consider arranging the two insets vertically instead of horizontally, as this layout could allow for greater magnification and a clearer visualization of the details.
10 - The Land Use and Land Cover analysis is well-presented and supported by clear mapping and quantitative data. However, the section would benefit from a stronger explanation of why the selected time intervals (2009, 2014, 2019, 2024) are particularly relevant to the research question, as well as a more explicit discussion of potential data limitations and accuracy issues. Strengthening the link between the observed spatial changes and their broader social and economic implications would further enhance the analytical depth of this part of the study.
11 - The questionnaire section provides a substantial amount of descriptive data, but its scientific rigor remains somewhat limited. The justification for the chosen questions and their alignment with the research objectives is not sufficiently articulated, and the balance between open- and closed-ended items could be better explained. Moreover, the process of distribution (e.g., via QR codes) raises concerns about potential sampling bias, particularly regarding the representativeness of respondents beyond frequent visitors to Al-Mouj. To strengthen this part, the authors should more clearly address sampling strategy, validity, and reliability, while also discussing how potential biases may affect the interpretation of results.
12 - The authors may wish to consider moving the tables related to the questionnaire to an annex. This adjustment would help streamline the main text, avoiding abrupt interruptions in the flow of reading while still making the material available for reference.
13 - The Results and Discussion section appears relatively underdeveloped in proportion to the overall paper. Given that this is where the core substance of the research lies, I strongly encourage the authors to expand and deepen this part. In particular, the discussion should place greater emphasis on interpreting the findings, highlighting their practical applications, and clarifying the broader social and economic implications. A more critical and reflective engagement with the results would considerably strengthen the contribution and impact of the study.

Author Response
Comments 1:
1 - I recommend that the authors consider adopting a more engaging title. A clearer and catchier formulation could better capture readers’ attention, enhance the paper’s visibility, and more effectively highlight the study’s main contribution.
Response 1:
Thank you for pointing this out. We agree with this comment. Therefore, A revised title has been proposed, considering the given constructive feedback.
Comments 2:
2 - I suggest that the authors place greater emphasis on the applications and implications of their research in the final part of the abstract, as this would strengthen the overall impact and relevance of the study.
Response 2:
Agree. The final part of the abstract has been modified to highlight the applications and implications of the research. (page 1, lines 31-34)
Comments 3:
3 - I recommend that the authors limit the number of keywords to a maximum of five and select them carefully, as these terms play a crucial role in indexing the paper in major academic search engines.
Response 3:
Agree. The Keywords have been reduced and are being focused on more. (page 1, lines 35-36)
Comments 4:
4 - The introduction provides a broad overview of the historical and conceptual foundations of sustainability and urban planning, but at times it reads more like a general textbook summary than a targeted framing of the research problem. While the references to the Brundtland Report, Howard’s “Garden City,” and Perry’s neighborhood unit are useful, the section would benefit from a clearer linkage between these classical theories and the specific context of sustainable neighborhoods in Oman. Furthermore, some parts could be streamlined to avoid redundancy and make space for a sharper articulation of the research gap. Highlighting more recent and internationally relevant literature on neighborhood sustainability would also strengthen the positioning of the study within current academic debates.
Response 4:
Agree. The introduction has been rewritten to clearly establish the connection between sustainable neighborhoods and Oman. Additionally, some sections have been streamlined, and recent, internationally relevant literature on neighborhood sustainability has been incorporated. (page 1, lines 39 - page 3, lines 127)
Comments 5:
5 - In the introduction, where authros state "[...] In recent years, awareness of sustainability issues has increased, as sustainability encompasses various aspects of resource use and society [...]", to better contextualize this sentence in a more recent and internationa literature scenario, the authors shoul cite here the following and very recent paper which addresses the various facets of sustainability in society:
-Falegnami, A., Romano, E. and Tomassi, A., 2024. The emergence of the GreenSCENT competence framework: A constructivist approach: The GreenSCENT theory. In The european green deal in education (pp. 204-216). Routledge. DOI: 10.4324/9781003492597-17
Response 5:
Thank you for pointing out this valuable book chapter, however, after careful review of it, the mentioned book chapter is quite far from the research topic and the highlighted sentence.
Comments 6:
6 - The authors should explicitly state whether the literature review was conducted according to a recognized systematic review protocol (such as PRISMA or ProKnow-C), or whether it should rather be considered a scoping review. Clarifying this aspect would improve the transparency and methodological rigor of the paper. In either case, the authors are encouraged to provide essential details of their search strategy, including the exact query used, the database(s) consulted, and the inclusion and exclusion criteria applied. Such information would not only allow readers to better assess the robustness of the review, but also facilitate reproducibility and increase the credibility of the research process.
Response 6:
The literature review was carried out following a systematic review process. At the start of the literature review section, the specific query used, the databases consulted, and the inclusion and exclusion criteria were outlined. (page 3, lines 129-137)
Comments 7:
7 - The methodology is generally well-structured, combining quantitative and qualitative approaches and employing recognized analytical tools. However, the section would benefit from greater clarity and detail in some areas, for instance, in explaining the rationale for selecting specific data collection methods, and in justifying how the sample size ensures representativeness of the studied population. Moreover, while multiple techniques (GIS, SPSS, NVivo) are mentioned, the link between each method and the corresponding research objective could be articulated more explicitly, thereby reinforcing the methodological rigor and transparency of the study.
Response 7:
The required clarification regarding the rationale of the selected data collection methods, justifying how ensuring the sample size represents the studied population, and the link between the selected techniques (GIS, SPSS, NVivo) and the research objectives has been articulated. (page 6, lines 253-281)
Comments 8:
8 - The case study paragraph is clearly described and provides useful contextual information about Al-Mouj and its surrounding area. Nonetheless, the section could be strengthened by clarifying the criteria that guided the choice of this specific case, and by better linking its characteristics to the research objectives. A more critical reflection on the representativeness and possible limitations of focusing on a single case would also enhance the robustness of the study.
Response 8:
The concern regarding the case study has been addressed and highlighted. (page 8, lines 333-354)
Comments 9:
9 - Figure 1 is of rather poor quality, and I kindly ask the authors to provide a higher-resolution version. In addition, they might consider arranging the two insets vertically instead of horizontally, as this layout could allow for greater magnification and a clearer visualization of the details.
Response 9:
Figure 1 has been resubmitted with a higher resolution and arranged vertically as recommended.
Comments 10:
10 - The Land Use and Land Cover analysis is well-presented and supported by clear mapping and quantitative data. However, the section would benefit from a stronger explanation of why the selected time intervals (2009, 2014, 2019, 2024) are particularly relevant to the research question, as well as a more explicit discussion of potential data limitations and accuracy issues. Strengthening the link between the observed spatial changes and their broader social and economic implications would further enhance the analytical depth of this part of the study.
Response 10:
The concern regarding the selected years for analysis and the required explanation has been addressed. (page 9, lines 360-361) and (page 11, lines 386-402)
Comments 11:
11 - The questionnaire section provides a substantial amount of descriptive data, but its scientific rigor remains somewhat limited. The justification for the chosen questions and their alignment with the research objectives is not sufficiently articulated, and the balance between open- and closed-ended items could be better explained. Moreover, the process of distribution (e.g., via QR codes) raises concerns about potential sampling bias, particularly regarding the representativeness of respondents beyond frequent visitors to Al-Mouj. To strengthen this part, the authors should more clearly address sampling strategy, validity, and reliability, while also discussing how potential biases may affect the interpretation of results.
Response 11:
Justification for the chosen questions and their alignment with the research objectives has been provided, and the balance between open- and closed-ended items has been explained. The process of distributing the QR has been clarified, along with how bias has been avoided. (page 11, line 405- page 12, line 438).
Comments 12:
12 - The authors may wish to consider moving the tables related to the questionnaire to an annex. This adjustment would help streamline the main text, avoiding abrupt interruptions in the flow of reading while still making the material available for reference.
Response 12:
The tables have been moved to the annex as recommended to help streamline the main text. (page 19, line 661 and page 19, lines 662).
Comments 13:
13 - The Results and Discussion section appears relatively underdeveloped in proportion to the overall paper. Given that this is where the core substance of the research lies, I strongly encourage the authors to expand and deepen this part. In particular, the discussion should place greater emphasis on interpreting the findings, highlighting their practical applications, and clarifying the broader social and economic implications. A more critical and reflective engagement with the results would considerably strengthen the contribution and impact of the study.
Response 13:
The results have been rewritten with more critical and reflective engagement and highlighting the practical applications. (page 15, line 511- page 18, line 629).
- Response to Comments on the Quality of English Language
Point 1: The English is fine and does not require any improvement
Response 1: Thank you
- Additional clarifications
Thank you for the valuable comments.
Please note that some parts of the Introduction and Literature Review sections have been revised to meet the Editorial requirements and comments.

Reviewer 2 Report
Comments and Suggestions for Authors
The manuscript explores an important and increasingly relevant topic within the field of urban sustainability by examining the social and economic impacts of a sustainable neighborhood, Al-Mouj, on its adjacent area, Al-Mawaleh North, in Muscat, Oman. It is commendable that the study aims to bridge a gap in regional literature, particularly within the context of Oman’s urban transformation under Vision 2040 and the Oman National Spatial Strategy. The focus on assessing how sustainable urban development influences surrounding communities is both timely and necessary, especially in the Global South where such empirical assessments remain limited. However, while the topic is relevant and the case study potentially valuable, the manuscript requires major revisions to enhance its academic rigor, methodological transparency, and overall coherence.
The title of the manuscript is broadly suitable, although it could benefit from a more concise and impactful formulation. Consider simplifying the phrasing while still capturing the core aim of the study. For example, “Assessing the Socio-Economic Impact of Sustainable Urban Development: The Case of Al-Mouj, Muscat” might be more effective and professional. Avoid unnecessary repetition of terms such as "investigating" or "case study" when they are implied within the structure and scope of the paper.
The introduction effectively outlines the national policy context and the strategic importance of sustainable neighborhoods in Oman’s development agenda. However, the conceptual framework is not sufficiently developed. There is limited engagement with current international literature on sustainable neighborhood development, social well-being, and economic resilience. Furthermore, the theoretical foundations for linking sustainability features with socio-economic outcomes should be better articulated. Currently, the manuscript relies heavily on descriptive framing without critically situating the study within broader academic debates.
The literature review section is notably outdated. Many of the references cited are prior to 2023, with some significantly older. This weakens the manuscript’s credibility and relevance, especially considering the rapidly evolving nature of sustainability discourses. To strengthen the foundation of the study, the authors should incorporate more recent and high-quality academic sources that reflect the latest developments in sustainable urbanism, socio-economic impact assessment, and the Middle Eastern urban context. This would not only enhance the study's relevance but also support a more refined articulation of its objectives and theoretical positioning.
The research design demonstrates a commendable effort to use mixed methods, including surveys, interviews, spatial analysis, and ethnographic approaches. However, the methodology section lacks sufficient detail for replication and proper assessment. More information is needed regarding how the 515 participants were selected, what sampling strategy was used, and how demographic balance was ensured. The integration of qualitative and quantitative data is not well explained. How the ethnographic findings were synthesized with statistical and spatial analyses should be made more transparent. In its current form, the methodological narrative reads as fragmented and overly broad, with multiple techniques mentioned but not cohesively linked.
The case study of Al-Mouj is inherently interesting and offers a potentially rich site for empirical inquiry, yet the manuscript does not fully capitalize on this potential. There is insufficient critical reflection on why Al-Mouj was selected, whether it is representative or exceptional, and how its impacts were traced or measured over time. Similarly, the development changes in Al-Mawaleh North are described in terms of percentages and land use change, but the analytical framework behind these measurements is not explained in sufficient depth. Claims such as a 135.7 percent increase in development between 2009 and 2024 require supporting data and clear methodological explanation to be credible.
The findings are generally presented in a descriptive manner, often stating that Al-Mouj had a positive influence on various social and economic factors without sufficient nuance or critical analysis. While the emphasis on improved mental and physical health, increased economic activity, and higher standards of living is welcome, it would be helpful to explore any negative externalities, unequal benefits, or issues of displacement, affordability, or gentrification. A more balanced discussion of the complexities of sustainable urban development would enhance the study’s value and depth.
The policy implications and conclusions are relatively sound but remain general. The manuscript would benefit from clearer, more actionable recommendations for urban planners, policy makers, and developers. These should be directly linked to the study’s findings and contextualized within the planning and governance realities of Oman. Furthermore, the implications for scaling such sustainable models across other parts of the country or the wider region should be briefly explored.
In terms of structure and presentation, the manuscript requires editorial improvements. Transitions between sections are often abrupt, and there is repetition in the text that could be streamlined. Key terms such as “sustainable neighborhood,” “mixed-use development,” and “well-being” should be clearly defined early on and used consistently throughout the manuscript. Figures, tables, or maps, if included, should be directly referenced and explained in the main text. If not present, the addition of visuals showing land use change or survey responses would greatly improve the clarity and engagement of the manuscript.
In summary, the manuscript tackles an important and underexplored topic in the context of sustainable urban development in the Middle East. However, to fulfill its potential and make a meaningful academic contribution, the paper must undergo substantial revisions. These include updating and expanding the literature base, clarifying the methodology, deepening the analysis of findings, and enhancing both structural coherence and theoretical engagement. With these improvements, the study could provide valuable insights for scholars, planners, and policy makers concerned with sustainable urbanism and community-centered development in the region.
Author Response
Comments 1:
The title of the manuscript is broadly suitable, although it could benefit from a more concise and impactful formulation. Consider simplifying the phrasing while still capturing the core aim of the study. For example, “Assessing the Socio-Economic Impact of Sustainable Urban Development: The Case of Al-Mouj, Muscat” might be more effective and professional. Avoid unnecessary repetition of terms such as "investigating" or "case study" when they are implied within the structure and scope of the paper.
Response 1:
Thank you for pointing this out. We agree with this comment. Therefore, A revised title has been proposed, considering the given constructive feedback.
Comments 2:
The introduction effectively outlines the national policy context and the strategic importance of sustainable neighborhoods in Oman’s development agenda. However, the conceptual framework is not sufficiently developed. There is limited engagement with current international literature on sustainable neighborhood development, social well-being, and economic resilience. Furthermore, the theoretical foundations for linking sustainability features with socio-economic outcomes should be better articulated. Currently, the manuscript relies heavily on descriptive framing without critically situating the study within broader academic debates.
Response 2:
The introduction has been rewritten to sufficiently develop the conceptual framework. Additionally, some sections have been streamlined, and recent, internationally relevant literature on neighborhood sustainability has been incorporated with more critical analysis. (page 1, lines 39 - page 3, lines 127)
Comments 3:
The literature review section is notably outdated. Many of the references cited are prior to 2023, with some significantly older. This weakens the manuscript’s credibility and relevance, especially considering the rapidly evolving nature of sustainability discourses. To strengthen the foundation of the study, the authors should incorporate more recent and high-quality academic sources that reflect the latest developments in sustainable urbanism, socio-economic impact assessment, and the Middle Eastern urban context. This would not only enhance the study's relevance but also support a more refined articulation of its objectives and theoretical positioning.
Response 3:
The literature review was conducted using a systematic process. At the beginning of the section, the specific query, the databases consulted, and the inclusion and exclusion criteria were outlined. It was based on tracking the development of literature regarding its social and economic impacts to establish appropriate criteria for the research, which is why it may include older sources. The manuscript included two papers from 2024 and three from 2025, but more references from 2025 related to the research have been added as they appeared. High-quality academic sources relevant to the research topic were considered. (page 3, lines 129-137)
Comments 4:
The research design demonstrates a commendable effort to use mixed methods, including surveys, interviews, spatial analysis, and ethnographic approaches. However, the methodology section lacks sufficient detail for replication and proper assessment. More information is needed regarding how the 515 participants were selected, what sampling strategy was used, and how demographic balance was ensured. The integration of qualitative and quantitative data is not well explained. How the ethnographic findings were synthesized with statistical and spatial analyses should be made more transparent. In its current form, the methodological narrative reads as fragmented and overly broad, with multiple techniques mentioned but not cohesively linked.
Response 4:
The integration of qualitative and quantitative data is explained more clearly, with a link to the research objectives. The methods of selecting the participants and ensuring demographic balance are also clarified in the methodology section and the questionnaire section. (page 6, lines 253-281)
Comments 5:
The case study of Al-Mouj is inherently interesting and offers a potentially rich site for empirical inquiry, yet the manuscript does not fully capitalize on this potential. There is insufficient critical reflection on why Al-Mouj was selected, whether it is representative or exceptional, and how its impacts were traced or measured over time. Similarly, the development changes in Al-Mawaleh North are described in terms of percentages and land use change, but the analytical framework behind these measurements is not explained in sufficient depth. Claims such as a 135.7 percent increase in development between 2009 and 2024 require supporting data and clear methodological explanation to be credible.
Response 5:
The reasons for selecting the case study and how the impacts were tracked have been clarified further. The analytical framework behind the measurements has also been explained in more depth. (page 8, lines 333-354)
Comments 6:
The findings are generally presented in a descriptive manner, often stating that Al-Mouj had a positive influence on various social and economic factors without sufficient nuance or critical analysis. While the emphasis on improved mental and physical health, increased economic activity, and higher standards of living is welcome, it would be helpful to explore any negative externalities, unequal benefits, or issues of displacement, affordability, or gentrification. A more balanced discussion of the complexities of sustainable urban development would enhance the study’s value and depth.
Response 6:
The findings have been rewritten with more critical analysis and some demonstration of some negative aspects that require some intervention. (page 15, line 512- page 17, line 590).
Comments 7:
The policy implications and conclusions are relatively sound but remain general. The manuscript would benefit from clearer, more actionable recommendations for urban planners, policy makers, and developers. These should be directly linked to the study’s findings and contextualized within the planning and governance realities of Oman. Furthermore, the implications for scaling such sustainable models across other parts of the country or the wider region should be briefly explored.
Response 7:
The conclusion has been rewritten to include more actionable recommendations for urban planners, policy makers, and developers. These suggestions have been connected to the realities of Oman and how they can be applied across other parts of the country. (page 17, line 591- page 18, line 629).
Comments 8:
In terms of structure and presentation, the manuscript requires editorial improvements. Transitions between sections are often abrupt, and there is repetition in the text that could be streamlined. Key terms such as “sustainable neighborhood,” “mixed-use development,” and “well-being” should be clearly defined early on and used consistently throughout the manuscript. Figures, tables, or maps, if included, should be directly referenced and explained in the main text. If not present, the addition of visuals showing land use change or survey responses would greatly improve the clarity and engagement of the manuscript.
Response 8:
The transition between sections has been smoothed to prevent abruptness, and repetitions in the text have been revised. The main text explains figures, tables, maps, and visuals showing land use changes, survey responses are already included. The key terms mentioned have been defined throughout the manuscript.
Comments 9:
In summary, the manuscript tackles an important and underexplored topic in the context of sustainable urban development in the Middle East. However, to fulfill its potential and make a meaningful academic contribution, the paper must undergo substantial revisions. These include updating and expanding the literature base, clarifying the methodology, deepening the analysis of findings, and enhancing both structural coherence and theoretical engagement. With these improvements, the study could provide valuable insights for scholars, planners, and policymakers concerned with sustainable urbanism and community-centered development in the region.
Response 9:
Thank you so much for your constructive feedback, which really enhanced the manuscript and addressed any missing points.
- Response to Comments on the Quality of English Language
Point 1: The English is fine and does not require any improvement
Response 1: Thank you
- Additional clarifications
Thank you for the valuable comments.
Please note that some parts of the Introduction and Literature Review sections have been revised to meet the Editorial requirements and comments.

Round 2
Reviewer 1 Report
Comments and Suggestions for Authors
The authors have addressed the comments.
Reviewer 2 Report
Comments and Suggestions for Authors
I have reviewed the revised version of the manuscript and would like to commend the authors for the substantial improvements made. The authors have addressed most of my previous comments and queries in a thorough and thoughtful manner. The revisions have notably strengthened the clarity, structure, and overall quality of the manuscript.